# VEGF Overexpression Significantly Increases Nanoparticle-Mediated siRNA Delivery and Target-Gene Downregulation

**DOI:** 10.3390/pharmaceutics14061260

**Published:** 2022-06-14

**Authors:** Shanshan Tan, Zhihang Chen, Yelena Mironchik, Noriko Mori, Marie-France Penet, Ge Si, Balaji Krishnamachary, Zaver M. Bhujwalla

**Affiliations:** 1Division of Cancer Imaging Research, The Russell H Morgan Department of Radiology and Radiological Science, Baltimore, MD 21205, USA; stan32@jhmi.edu (S.T.); zchen19@jhu.edu (Z.C.); ymironc1@jhmi.edu (Y.M.); nmori3@jhmi.edu (N.M.); mpenet2@jhmi.edu (M.-F.P.); gsi1@jhu.edu (G.S.); bkrishn1@jhmi.edu (B.K.); 2Sidney Kimmel Comprehensive Cancer Center, Baltimore, MD 21205, USA; 3Department of Chemical and Biomolecular Engineering, The Johns Hopkins University, 3400 N. Charles Street, Baltimore, MD 21218, USA; 4Department of Radiation Oncology and Molecular Radiation Sciences, The Johns Hopkins University School of Medicine, Baltimore, MD 21205, USA

**Keywords:** choline kinase, nanoparticles, siRNA delivery, triple negative breast cancer, vascularization

## Abstract

The availability of nanoparticles (NPs) to deliver small interfering RNA (siRNA) has significantly expanded the specificity and range of ‘druggable’ targets for precision medicine in cancer. This is especially important for cancers such as triple negative breast cancer (TNBC) for which there are no targeted treatments. Our purpose here was to understand the role of tumor vasculature and vascular endothelial growth factor (VEGF) overexpression in a TNBC xenograft in improving the delivery and function of siRNA NPs using in vivo as well as ex vivo imaging. We used triple negative MDA-MB-231 human breast cancer xenografts derived from cells engineered to overexpress VEGF to understand the role of VEGF and vascularization in NP delivery and function. We used polyethylene glycol (PEG) conjugated polyethylenimine (PEI) NPs to deliver siRNA that downregulates choline kinase alpha (Chkα), an enzyme that is associated with malignant transformation and tumor progression. Because Chkα converts choline to phosphocholine, effective delivery of Chkα siRNA NPs resulted in functional changes of a significant decrease in phosphocholine and total choline that was detected with ^1^H magnetic resonance spectroscopy (MRS). We observed a significant increase in NP delivery and a significant decrease in Chkα and phosphocholine in VEGF overexpressing xenografts. Our results demonstrated the importance of tumor vascularization in achieving effective siRNA delivery and downregulation of the target gene Chkα and its function.

## 1. Introduction

There has been a major resurgence of interest in using small interfering RNA (siRNA) to silence genes in the treatment of diseases [1]. Effective delivery and cellular uptake of siRNA are major challenges in the applications of siRNA therapy [2] that become even more challenging in cancer because of the abnormal vasculature of tumors. Theranostic nanoparticles (NPs) that integrate siRNA delivery with imaging reporters provide excellent opportunities for visualization of siRNA delivery and uptake. Tumor vasculature is typically disorganized with immature capillaries that are leaky and tortuous and cannot provide sufficient oxygen and nutrients to cells within the tumor, resulting in hypoxia, acidic pH, and necrosis. Studies directly relating vascularization to the visualization of siRNA NP delivery and functional outcome are required to understand how these NPs navigate through the tumor microenvironment (TME), and to understand the impact of the TME on siRNA delivery and function [3]. Previous studies have shown the importance of effective delivery in the downregulation of the target gene [4], but the functional outcome of the downregulation of the target gene has not been verified. Our purpose here was to evaluate the effect of increased vascularization on siRNA NP delivery and the functional outcome of the target gene downregulation.

Breast cancer is one of the most commonly diagnosed cancers, and the second leading cause of cancer-related deaths among women in the United States [5]. Triple negative breast cancer (TNBC) is an aggressive subtype of breast cancer that lacks the expression of human epidermal growth factor receptor 2 (HER2), progesterone receptor (PR), and estrogen receptor (ER). TNBC accounts for 10~15% of all breast cancer cases [6,7] and is associated with poor prognosis [8,9]. Because of the lack of conventional targets in TNBC, siRNA therapy presents a novel opportunity for treating TNBC. Among potential targets for TNBC, choline kinase α (Chkα) is an attractive candidate. As an enzyme that catalyzes the conversion of choline to phosphocholine (PC), Chkα is associated with malignant transformation and tumor progression [10,11]. Pharmacological inhibitors of Chkα have shown some antitumor activity [12,13]. However, a novel potent and selective small-molecule Chkα inhibitor, V-11-0711, which acts by significantly inhibiting the catalytic activity of Chkα [14], reduced PC but did not cause cell death [14,15]. Downregulation of the Chkα protein, on the other hand, consistently decreased cell survival [15]. These studies highlighted the potential use of Chkα siRNA in TNBC treatment, underscoring the need to identify factors that can improve the effectiveness of Chkα siRNA such as vascular delivery.

Here we used optically labeled polyethylene glycol (PEG)-polyethylenimine (PEI) NPs to visualize the delivery of Chkα siRNA to TNBC MDA-MB-231 xenografts with or without vascular endothelial growth factor (VEGF) overexpressed. VEGF is a potent angiogenic and permeability factor that is expressed by cancer cells to establish angiogenesis [16]. It is frequently upregulated in breast cancer tissue [17,18,19], and it is also transcriptionally regulated by hypoxia [20]. We have previously detected significantly higher vascular volume in MDA-MB-231 tumors derived from cells engineered to overexpress VEGF [21]. The use of TNBC xenografts with and without VEGF overexpression allowed us to evaluate the role of vascular delivery in target gene downregulation. Because hypoxia occurs in TNBC [22], we also evaluated the effects of hypoxia on the ability of siRNA to downregulate the target gene. The use of Chkα siRNA provided the added advantage of evaluating the functional outcome of Chkα downregulation through the reduction in PC and total choline that can be detected with ^1^H magnetic resonance spectroscopy (MRS), allowing an evaluation of the functional impact of the siRNA within the heterogeneous TME.

## 2. Materials and Methods

### 2.1. Cell Lines and Tumors

MDA-MB-231 human breast cancer cells (ATCC, Manassas, VA, USA) were engineered to stably express VEGF as previously described [21]. Briefly, cDNA for VEGF165 (Genentech Inc, South San Francisco, CA, USA) was cloned into the eukaryotic expression vector pCR3.1 under the control of a constitutive CMV promoter. MDA-MB-231 cells were cultured using RPMI 1640 medium (Sigma^®^, Saint Louis, MO, USA) supplemented with 10% fetal bovine Serum (FBS, Sigma^®^, Saint Louis, MO, USA). MDA-MB-231 VEGF cells were cultured with the same medium with 400 μg/mL of G418 Sulfate (Corning™, Glendale, AZ, USA). Expression of VEGF was routinely checked by RT-PCR.

Tumors were obtained by inoculating 10^6^ MDA-MB-231 wild-type (231 WT) or MDA-MB-231 VEGF (231 VEGF) cells, orthotopically, in female severe combined immunodeficient (SCID) mice. Tumors reached volumes of 300–400 mm^3^ within 4–6 weeks after inoculation, at which point the animals were used for study. All animal studies were done in compliance with a protocol approved by the Animal Care and Use Committee of the Johns Hopkins University School of Medicine.

### 2.2. Preparation and Characterization of Chkα siRNA PEG-PEI NPs

Branched PEI (PEI, Aldrich^®^, Saint Louis, MO, USA; M_w_ 25,000) was used to form noncovalent inter-polyelectrolyte NPs. Polyethylene glycol 2000 (PEG 2000) NHS ester (NanoCS, Natick, MA, USA) was reacted with PEI 25k at a 15:1 molar ratio. Excess PEG was removed by MilliporeSigma™ Amicon™ Ultra Centrifugal Filter Units (St. Louis, MO, USA) with a molecular weight cutoff of 10,000 Da. ^1^H NMR spectroscopy (Bruker Avance III 500 MHz NMR spectrometer, Bruker, Billerica, MA, USA) demonstrated that the PEG reacted with PEI at an average molar ratio of 9:1 (Appendix A). Cy5.5 NHS ester (Click Chemistry Tools, Scottsdale, AZ, USA) was reacted with PEGylated PEI at a 1:1 molar ratio in phosphate buffered saline (PBS, pH = 7.4) to provide an optical imaging reporter for imaging the PEG-PEI NPs.

Modified PEG-PEI was mixed and incubated at ambient temperature with Chkα siRNA in reduced serum for cell studies, or in PBS for in vivo studies. The N/P ratio, the ratio of the number of nitrogen atoms in one molecule of PEG-PEI to the number of phosphor atoms in one molecule of Chkα siRNA, was 15. PEG-PEI NPs were prepared by mixing modified PEI with Chkα siRNA approximately 30 min to one hour prior to each experiment. The characterization of the PEG-PEI NPs was performed by transmission electron microscopy (TEM, Hitachi 7600, Tokyo, Japan) and by dynamic light scattering (DLS, Malvern Zetasizer Nano ZS90, Malvern, United Kingdom). A representative TEM image is presented in Appendix A, with the histogram of the NP size distribution presented in Appendix A. The mean diameter as estimated by TEM was 15.4 nm. The hydrodynamic diameter of the PEG-PEI NPs was on average 108 nm with a polydisperse index (PdI) of 0.336 as shown in Appendix A. The zeta-potential of the PEG-PEI siRNA complex was approximately 7.0 mV as previously estimated [23]. While we have not performed characterization of PEG-PEI NP stability in this study, previous studies have demonstrated the stability of PEG-PEI NPs in serum [24].

### 2.3. In Vitro RNA Interference

The in vitro efficiency of RNA interference by Chkα siRNA PEG-PEI NPs was evaluated in 231 WT and 231 VEGF cells. All siRNA used in the study were obtained from Dharmacon™ (Lafayette, CO, USA). A previously validated Chkα siRNA sequence [25], 5′-CAUGCUGUUCCAGUGCUCC-3′, was designed using the Thermo Scientific siRNA Design Center (Thermo Scientific, Rockford, IL, USA). Approximately 0.4 × 10^6^ cells were seeded in 60 mm culture dishes and cultured overnight. The following day, 0.2 nmol of Chkα siRNA was mixed with PEG-PEI at an N/P ratio of 15 in 50 µL of Opti-MEM^TM^ reduced serum medium for 30 min before being added to the petri dish. The siRNA concentration was 100 nM. Untreated cells served as a negative control. Cells treated with an identical amount of Chkα siRNA mixed with DharmaFECT 4 as the transfection reagent were used as a positive control. After 24 h of incubation, cells were collected and the Chkα mRNA level of each group evaluated by RT-PCR.

### 2.4. Immunohistochemistry Staining

CD31 immunohistochemistry (IHC) staining was used to identify vasculature in tumor sections. Both 231 WT and 231 VEGF tumors were harvested, fixed in formalin, and embedded with paraffin. Tumor tissue slides were sectioned at 5 µm thickness, then dewaxed and rehydrated. Antigen retrieval was achieved by boiling the slides in pre-warmed citrate buffer, pH 6.0 solution for 20 min. Peroxidase blocking and serum free protein blocking were performed on slides prior to overnight incubation at 4 °C with a rat monoclonal CD31 antibody (platelet endothelial cell adhesion molecule-1, PECAM-1 DIA 310, clone SZ31, Rat IgG2A, Dianova, Hamburg, Germany, 1:30 dilution). Horseradish peroxidase (HPR) conjugated secondary antibody (Vector Laboratories, Burlingame, CA, USA) was used to recognize the primary antibody. After incubation with secondary antibody for 1 h, DAB (3,3′-diaminobenzidine) chromogen was used to develop color, following which slides were counter stained with hematoxylin (Vector Laboratories, Burlingame, CA, USA).

Stained tissue slides were scanned with an Aperio ScanScope XT slide scanner (Aperio Technologies, Vista, CA, USA) and analyzed by Aperio Imagescope. The number of strongly positive pixels normalized to the total number of pixels was obtained. Analyses were performed using the entire histological section.

### 2.5. Optical Imaging

PEG-PEI NPs labeled with Cy5.5 dye were imaged with a Pearl^®^ Trilogy Small Animal Imaging System (LI-COR, Lincoln, NE, USA). PEG-PEI NPs delivering Chkα siRNA were administered intravenously with two doses of 3 nmol of Chkα siRNA, N/P ratio 15 (0.1 mg of NP per dose), each given 24 h apart, once tumors were ~300–400 mm^3^. In vivo and ex vivo optical imaging were used to track NP delivery and distribution in mice with 231 WT tumors and 231 VEGF tumors at 24 h following injection of the second dose.

### 2.6. In Vivo ^1^H MR Spectroscopic Imaging and Ex Vivo High-Resolution ^1^H MRS

MR studies were performed on a 4.7 T Bruker scanner with a solenoidal coil placed around the tumor. Total choline was detected with ^1^H magnetic resonance spectroscopic imaging (MRSI) using a 2D chemical shift imaging (CSI) sequence before and after treating with 2 doses of Chkα siRNA NPs. Water-suppressed ^1^H MRSI was performed with the following parameters: slice thickness, 4 mm; water suppression, VAPOR; TR, 1059 ms; TE, 272 ms; number of scans, 4; and FOV, 1.6 cm × 1.6 cm as previously described [26].

High resolution ^1^H MRS was performed on tumor extracts obtained with dual phase extraction as previously described [27], with the following modification. Dual phase extraction was used on snap-frozen tumor tissue with methanol/chloroform/water (1.5/3/1) individually added and sonicated before separation of the aqueous and lipid phases. High-resolution ^1^H MR spectra were acquired on a Bruker Biospin Avance-III 750 MHz MR spectrometer (Bruker Biospin Billerica, MA, USA) using a 5 mm broadband inverse (BBI) probe. 600 μL of 1× phosphate buffered deuterated water (D_2_O) (90% D_2_O, 10% H_2_O) containing 3-(trimethylsilyl) propionic 2,2,3,3-d4 acid sodium salt (TSP), as an internal standard, was used to resuspend the dried aqueous phase extract for MRS analysis. Spectra of the aqueous phase with water suppression were acquired with pre-saturation using a single pulse sequence with the following parameters: spectral width of 15,495.87 Hz, data points of 64 K, 90° flip angle, relaxation delay of 10 s, acquisition time of 2.11 s, 64 scans with 8 dummy scans, receiver gain 256. MR spectra were processed using Bruker Topspin 4.1.3. Integrals of metabolite signals including PC, glycerophosphocholine (GPC), and choline (Cho) were measured and normalized to the weight of tissue and compared to the TSP standard to obtain relative concentrations in arbitrary units (A.U.).

### 2.7. RT-PCR and Immunoblotting

RNA from cells and snap-frozen tumor tissue was extracted by QIAshredder and RNeasy Mini kit (Qiagen, Valencia, CA, USA) following the manufacturer’s protocol. cDNA was synthesized by iScript cDNA synthesis kit (Bio-Rad, Hercules, CA, USA). A 10× dilution of cDNA sample and Chkα specific primers were used for quantitative real-time PCR analysis with an iCycler real-time PCR detection system (Bio-Rad). The relative Chkα gene expression to an endogenous control hypoxanthine phosphoribosyltransferase 1 (HPRT1) was calculated based on the delta delta Ct method, where relative gene expression was presented as 2^−ΔΔCt^.

Total protein from tumor tissue was extracted with a RIPA (Radio-immunoprecipitation assay) buffer with various protease and phosphatase inhibitors to prevent degradation of the protein [28]. For protein electrophoresis and immunoblotting analysis, 7.5% SDS-PAGE gel was used. A customized rabbit polyclonal primary antibody against Chkα, and an anti-GAPDH antibody (mouse monoclonal, Sigma, St. Louis, MO, USA) were used. Immunoblots were developed by a SuperSignal™ West Pico PLUS Chemiluminescent Substrate kit (ThermoFisher Scientific, Waltham, MA, USA) following the manufacturer’s instructions.

### 2.8. Statistics Analysis

Values represent mean ± SEM from three or more experiments, with experimental numbers provided in the corresponding figure legend. Statistical differences between different groups were evaluated by a two-tailed unpaired t-test using GraphPad Prism 5. A *p* value ≤ 0.05 was considered statistically significant.

## 3. Results

### 3.1. VEGF Overexpression Promotes Vascularization

The VEGF expression by 231 WT and 231 VEGF cells was evaluated by RT-PCR. 231 VEGF cells demonstrated a significantly higher level of VEGF mRNA expression compared to 231 WT cells (Figure 1A).

Representative immunostained tumor sections of the endothelial cell marker CD31 presented in Figure 1B show increased CD31 density in the 231 VEGF tumor. A summary of CD31 IHC data analysis presented in Figure 1C demonstrated that the strongly positive pixel fraction significantly increased from 1.27 ± 0.38 in 231 WT tumors to 4.60 ± 0.41 in 231 VEGF tumors. This significant increase in CD31 confirmed the increased angiogenesis induced by VEGF in these tumors.

### 3.2. Downregulation of Chkα in Cells Following Treatment with Chkα siRNA PEG-PEI NPs

Treatment of 231 WT or 231 VEGF cells with Chkα siRNA NPs resulted in a significant and comparable decrease in Chkα mRNA, as shown in Figure 2A,B. To directly determine the effects of hypoxia on RNA interference, we examined the siRNA-mediated downregulation of Chkα in 231 WT cells under normoxia or hypoxia using the transfection agent D-FECT 4 or the PEG-PEI NPs. As shown in Appendix A, hypoxia did not affect RNA interference, as the target gene was downregulated to the same extent under normoxia or hypoxia when using D-FECT 4 or the PEG-PEI NPs. Although D-FECT 4 was more effective at downregulation of Chkα than PEG-PEI NPs (0.25-fold or 75% reduction with D-FECT vs. 0.5-fold or 50% reduction with PEG-PEI NPs as compared to control), D-FECT cannot be used for in vivo siRNA delivery.

### 3.3. Chkα siRNA NP Biodistribution with In Vivo and Ex Vivo Optical Imaging

We performed in vivo and ex vivo optical imaging to determine the delivery and biodistribution of the PEG-PEI NPs. In vivo and ex vivo imaging of mice with 231 WT and 231 VEGF tumors was performed 24 h after the Chkα siRNA treatment that consisted of two doses of 3 nmol of Chkα siRNA delivered 24 h apart. Following in vivo imaging, mice were euthanized to perform ex vivo imaging of excised tissues.

As shown in the representative in vivo images in Figure 3A, markedly higher fluorescence intensity was observed in the 231 VEGF tumor (right) compared to the 231 WT tumor (left). Representative ex vivo images in Figure 3B obtained from a 2 mm thick slice of excised tumor (bottom row) and muscle tissue (top row) confirmed the high intensity in the 231 VEGF tumor compared to the 231 WT tumor, reflecting increased delivery of the NPs in tumors with VEGF overexpressed. Quantitative analysis of the tumor to muscle fluorescence intensity ratio summarized in Figure 3C confirmed the almost five-fold increase in NPs in the 231 VEGF tumor group compared to the 231 WT group.

Representative images of the biodistribution of the NPs in the liver, kidney, lung, heart, and spleen are presented in Figure 4A. Tumor VEGF overexpression did not alter the uptake of NPs in these organs, as evident from the comparable organ intensities.

Results of the ex vivo organ imaging studies from all the mice are summarized in Figure 4B. Although NP accumulation in the spleen appeared to be higher in the WT compared to the VEGF tumor-bearing mouse in the representative image in Figure 4A, when averaged across all the mice in each group, there was no significant difference in spleen accumulation between the two groups. There were no significant differences between NP uptake in any imaged organs of mice bearing tumors with or without VEGF overexpression. In vivo and ex vivo Cy5.5 fluorescence images of NP uptake obtained in all the tumors and organs are presented in Appendix A for 231 WT tumor-bearing mice and in Appendix A for 231 VEGF tumor-bearing mice. A higher uptake of the NPs is evident in the 231 VEGF tumors compared to the 231 WT tumors.

### 3.4. Downregulation of Chkα and Decrease in Total Choline in MDA-MB-231-VEGF Tumors

We compared the fold decrease in Chkα mRNA with or without VEGF overexpression following two doses of siRNA NP administration, normalized to untreated control tumors. As shown in Figure 5A, Chkα mRNA levels were significantly lower in treated 231 VEGF tumors compared to treated 231 WT tumors.

A correlation plot between tumor to muscle fluorescence intensity of the VEGF group as an index of NP delivery, and mRNA fold change, showed the dependence of downregulation efficacy on NP delivery (Figure 5B). Immunoblots of tumor extracts (Figure 5C) confirmed the decrease in Chkα protein in the 231 VEGF tumors following siRNA NP treatment, whereas Chkα protein in treated 231 WT tumors remained comparable to protein levels in untreated 231 WT and VEGF tumors.

An example of the functional reduction in total choline (tCho) detected noninvasively in vivo with ^1^H MRSI following two doses of Chkα siRNA NPs in a 231 VEGF tumor is shown in Figure 5D. An overall reduction in tCho was identified, although heterogeneity of tCho reduction is evident in the post-treatment image.

### 3.5. Choline Metabolite Levels in Tumor Extracts

To further characterize the delivery-dependent functional effects of Chkα siRNA on choline metabolites, we analyzed GPC, PC, Cho, and total choline (Cho + PC + GPC) levels as detected in high-resolution ^1^H MR spectra of tumor extracts. Representative ^1^H MR spectra obtained from the aqueous phase of extracts of untreated (left) and treated (right) 231 WT (Figure 6A) and 231 VEGF (Figure 6B) tumors show the decrease in PC in the 231 VEGF treated tumor. Data summarized for 231 WT tumors in Figure 6C show that PC levels did not change significantly between the untreated and treated tumors (Figure 6C). In contrast, effective delivery of the NPs resulted in a significant decrease in PC levels in the treated 231 VEGF tumors compared to untreated 231 VEGF tumors (Figure 6D). GPC, Cho, and total choline in the 231 WT and the 231 VEGF tumor groups did not change significantly following siRNA NP treatment. Baseline levels of choline metabolites in untreated 231 WT and untreated 231 VEGF tumors were also compared. Although there was a trend towards lower Cho in untreated 231 VEGF tumors compared to untreated 231 WT tumors, no significant differences were identified in levels of GPC, PC, Cho, and total choline between untreated 231 VEGF and 231 WT tumors, confirming that VEGF overexpression did not alter the basal choline metabolites in these tumors (Appendix A).

## 4. Discussion

VEGF overexpressing MDA-MB-231 tumors clearly showed a significantly higher delivery of the NPs compared to wild-type tumors, although organ uptake was comparable in both groups. VEGF overexpression resulted in a significant increase in endothelial cells as identified by CD31 IHC, confirming our earlier observations that VEGF increased vascular volume in these tumors [21]. VEGF is also a potent permeability factor [29] that would have facilitated the extravasation of the NPs from the vasculature, although prior MRI characterization of permeability did not detect a significant increase in permeability [21]. The increased delivery of Chkα siRNA NPs in VEGF overexpressing tumors resulted in a significant decrease in Chkα as identified through molecular characterization of mRNA and protein. The functional impact of increased Chkα siRNA resulted in a significant decrease in PC as detected by ^1^H MRS, confirming a direct relationship between increased delivery and functional outcome in these tumors.

The ability to silence any gene of interest makes siRNA a powerful technology that significantly expands the targets available for precision therapy [30]. In addition, siRNA treatment does not result in genome modification, which is beneficial in terms of safety considerations. However, instability in blood and limited intracellular delivery have been barriers in the clinical applications of siRNA. As a result, there have been significant efforts to develop siRNA delivery systems that increase siRNA stability and cell penetration [31,32,33]. Effective siRNA encapsulation [34,35,36] and protection [37,38] are two the main strategies applied in developing delivery systems. Here we selected PEGylated PEI to deliver siRNA based on its ability to form noncovalent interpolyelectrolyte complexes with siRNA [39] that allowed stability and cell penetration [33,40]. We previously evaluated PEG-PEI cytotoxicity using an MTT assay in MDA-MB-231 cells treated for 4 days with varying concentrations of PEG-PEI and found that PEG-PEI NP concentrations of 0.3 µM or less did not reduce cell viability [23]. Based on the injection dose of 0.1 mg PEG-PEI in a 25 g mouse, we anticipate that the in vivo concentration of PEG-PEI of ~0.1 µM did not induce cytotoxicity over a 4-day period, by which time the complex would have been excreted. NPs can also react differently based on the protein corona of cells [41]. In our study, we used a pair of isogenic cancer cell lines with and without VEGF expression to investigate the effects of increasing tumor vascularity on NP delivery and downregulation of the target gene. Because we used isogenic cancer cells in SCID mice, we do not anticipate that plasma proteins would have been altered, but this effect should also be considered. Because of the size of the NPs used in this study, we anticipate that the NPs were cleared by the reticuloendothelial system (RES). Several studies have investigated the pharmacokinetics and clearance of PEG-PEI NPs [42]. These NPs are cleared by the RES, with degradation products that undergo renal and hepatobiliary elimination [43]. Our imaging studies showed the highest uptake in the kidney, followed by the liver, 24 h after two doses of the NPs. By 24 h, degradation of the NPs undergoes both renal and hepatobiliary elimination. The images suggest that there was more renal clearance at 24 h.

While several NPs that provide siRNA stability during circulation and cell penetration are available [33,40], the delivery and distribution of these NPs within the tumor is the next barrier regarding their translational use to downregulate target genes. For the PEG-PEI NPs used in this study, the in vivo and ex vivo optical imaging studies clearly identified the heterogeneity of NP distribution within the tumors, and the need for imaging to establish effective siRNA delivery. The heterogeneity of NP distribution was also consistent with the heterogeneity of tCho reduction as identified with in vivo ^1^H MRSI.

Cancer cells induce neovascularization by co-opting and remodeling existing vasculature [44], or by vasculogenesis, through the formation of de novo capillaries from endothelial progenitor cells [45,46]. Tumor angiogenesis arises from stress responses to low oxygen and nutrients resulting in the overproduction of pro-angiogenic cytokines, such as VEGF, by cancer cells and by stromal cells present in the tumor [47,48]. Increased concentrations of cytokines result in deregulation of the normal angiogenic cascade [49,50], resulting in a disorganized and chaotic vasculature. Immature, highly permeable blood vessels in tumors are leaky to macromolecules, have reduced blood flow, and are unable to support tumor oxygen and nutrient requirements of cancer cells, resulting in hypoxia and necrosis [51]. Because hypoxia is frequently observed in the TME of cancers, it was important to establish the ability of siRNA to downregulate the target gene in the presence of hypoxia. Our cell studies confirmed that hypoxia did not alter the effectiveness of siRNA in downregulating the target gene. However, because tumor hypoxic areas are associated with poor perfusion [52], siRNA NP delivery and consequently target gene downregulation will be less effective in hypoxic tumor regions.

Previous studies have identified the importance of Chkα as a target in breast cancer as well as in other cancer types [53]. As a result, pharmacological inhibitors of Chkα have been developed [54] that have also been evaluated in a Phase I clinical trial (NCT01215864). However, studies have also shown that Chkα acts as a chaperone protein [55,56] making it important to destabilize or downregulate the protein itself rather than its catalytic activity, a strategy where treatment with Chkα siRNA NPs is important. Studies with MDA-MB-231 xenografts treated with repeated doses of a lentiviral vector delivered systemically to transduce cancer cells in vivo with Chkα shRNA identified a significant reduction in cancer cell proliferation in treated tumors [57]. Our purpose here was to identify the effects of increased vascularization on Chkα siRNA delivery and the target-gene functional outcome.

In conclusion, the importance of vascularization in siRNA NP delivery and target-gene downregulation was demonstrated from the significant increase in Chkα siRNA NPs in 231 VEGF tumors, the significant reduction in Chkα mRNA and protein, and the significant reduction in PC in these tumors. Clearly, increased delivery resulted in greater downregulation of the target gene and its functional outcome. Even with increased vascularization, heterogeneity of delivery and functional outcome was evident. Our data demonstrate the feasibility using siRNA to downregulate target genes in vivo using NP delivery systems for future translational applications. Our data also highlight the importance of imaging to detect the delivery and the heterogeneity of siRNA NPs in tumors to predict the effectiveness of such treatments. Within the complex TME, vascularization appears to be a dominant factor in the outcome of treatment with siRNA NPs. Here we increased vascularization by overexpressing VEGF. While the increased vascularization effects of VEGF are well established, studies have also identified the role of VEGF in altering the extracellular matrix [58]. Future studies should investigate the role of VEGF-mediated changes in the extracellular matrix in contributing to the increased delivery and distribution of the Chkα siRNA NPs in these tumors. Strategies to improve delivery of NPs in tumors are important in the translational use of siRNA to achieve gene-specific downregulation.

## Figures and Tables

**Figure 1 pharmaceutics-14-01260-f001:**
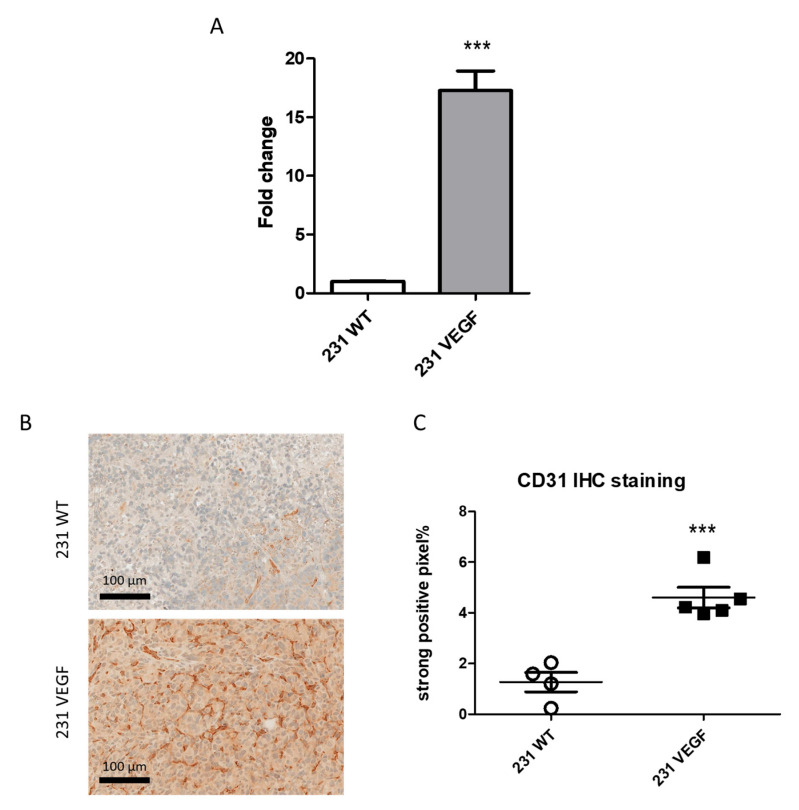
VEGF overexpression and increase in endothelial cells. (**A**) RT-PCR demonstrated a 17-fold increase in VEGF mRNA in 231 VEGF cells compared to 231 WT cells. Values represent mean ± SEM from 3 experiments, *** *p ≤* 0.001. (**B**) Representative CD31 immunostained sections from a 231 WT tumor (top) and 231 VEGF tumor (bottom), scale bar: 100 µm. (**C**) Comparison of percent CD31 strongly positive pixels in immunostained sections obtained from 231 WT (*n* = 4) and 231 VEGF (*n* = 5) tumors. Values represent mean ± SEM, *** *p ≤* 0.001.

**Figure 2 pharmaceutics-14-01260-f002:**
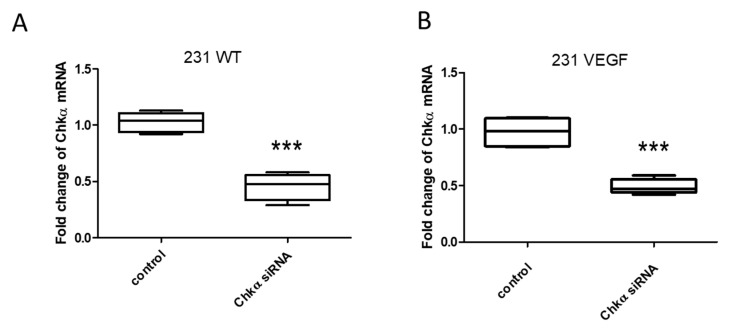
Downregulation of Chkα mRNA in 231 WT and 231 VEGF cells. (**A**) Relative fold reduction in Chkα mRNA in 231 WT cells (0.46 ± 0.06) following treatment with Chkα siRNA NPs. (**B**) Relative fold reduction in Chkα mRNA in 231 VEGF cells (0.49 ± 0.03) following treatment with the Chkα siRNA NPs. Values represent mean + SEM; *** *p* < 0.001. Relative fold changes were normalized to corresponding untreated cells used as controls; values are from 4 to 6 experiments; siRNA concentration in the medium was 100 nM.

**Figure 3 pharmaceutics-14-01260-f003:**
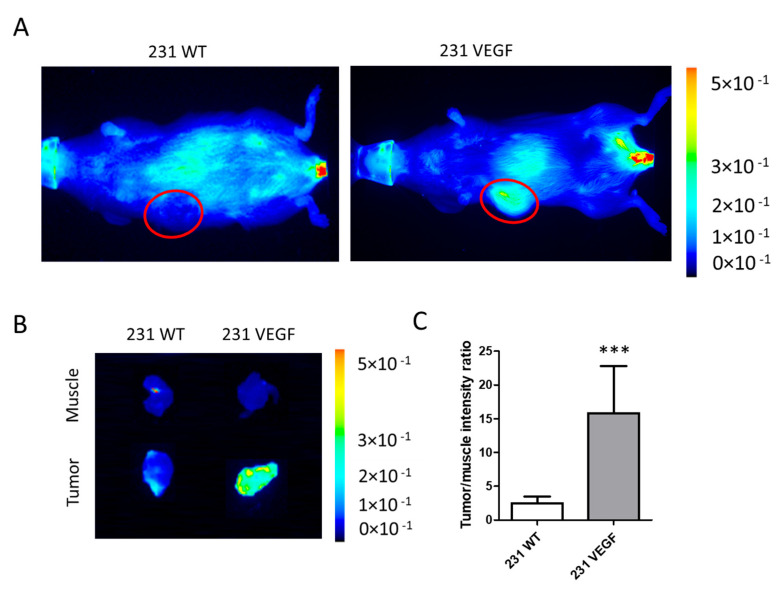
In vivo delivery of Chkα siRNA PEG-PEI NPs. (**A**) Representative in vivo Cy5.5 NIR fluorescence images of mice bearing 231 WT (**left**) and 231 VEGF (**right**) tumors 24 h after two doses of Chkα siRNA NPs. The tumor region is outlined with a red circle. A higher signal intensity was evident in the 231-VEGF tumor compared to the 231 WT tumor. (**B**) Corresponding representative ex vivo images of 2 mm thick fresh tissue slices from tumor and muscle tissue. Muscle tissue from a 231 WT tumor-bearing mouse (**top row**, **left**) displayed similar signal intensity as muscle tissue from a 231 VEGF tumor-bearing mouse (**top row**, **right**). In contrast, the 231 VEGF tumor slice (**bottom row**, **right**) exhibited appreciably higher intensity compared to the 231 WT tumor slice (**bottom row**, **left**). (**C**) Quantitative analysis of the ex vivo tumor/muscle intensity ratio from 231 WT and 231 VEGF tumors showed a significant increase in 231 VEGF tumors compared to 231 WT tumors. Values represent mean + SEM, *** *p* < 0.001, *n* = 6 for each group. Each mouse received 3 nmol of siRNA per injection.

**Figure 4 pharmaceutics-14-01260-f004:**
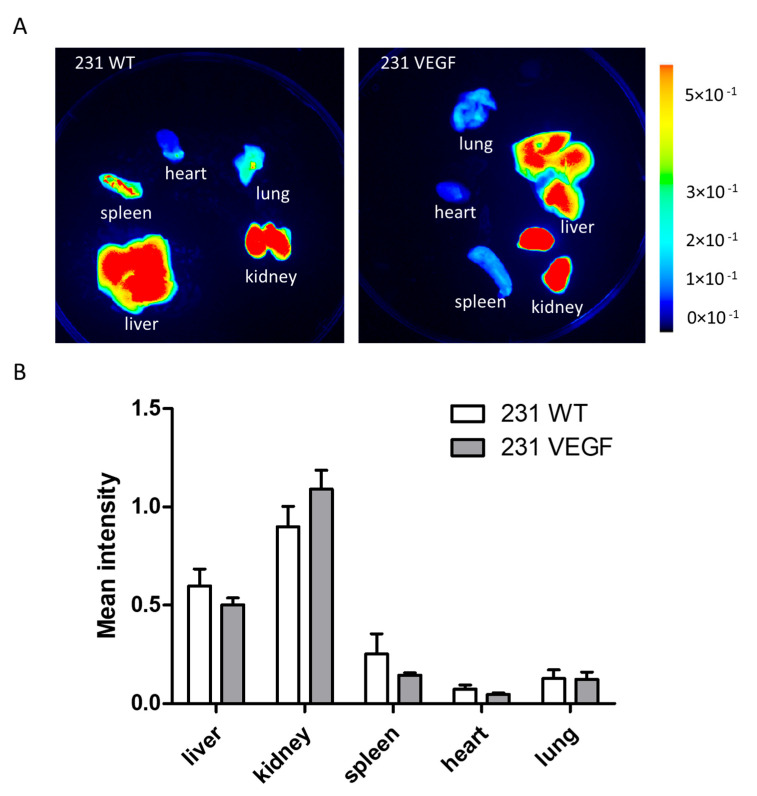
Ex vivo organ biodistribution of Chkα siRNA NPs. (**A**) Representative ex vivo images of fluorescence intensity in organs from a mouse bearing a 231 WT tumor (**left**) and a 231 VEGF tumor (**right**), 24 h after two doses of Chkα siRNA NPs. (**B**) Quantitative image analysis of ex vivo organ tissues from mice bearing 231 WT and 231 VEGF tumors. No significant difference in fluorescence intensity was observed in the organs obtained from mice with 231 WT tumors compared to mice with 231 VEGF tumors. Values represent mean ± SEM, *n* = 6 for each group.

**Figure 5 pharmaceutics-14-01260-f005:**
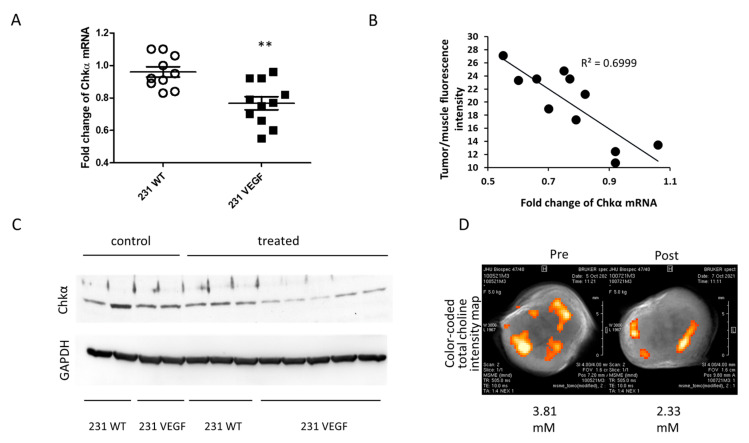
Improved downregulation of Chkα mRNA and protein in 231 VEGF tumors. (**A**) Fold change of Chkα mRNA levels in 231 WT (*n* = 10) and 231 VEGF (*n* = 11) tumors; a significant reduction in Chkα mRNA in 231 VEGF compared to 231 WT tumors was observed. Values normalized to untreated tumors represent Mean ± SEM, ** *p* < 0.01. (**B**) Correlation between Chkα mRNA levels in 231 VEGF tumor tissue and the tumor/muscle fluorescence intensity ratio following Chkα NP administration. (**C**) Immunoblots of 231 WT and 231 VEGF tumors with and without Chkα siRNA NP administration. Following Chkα siRNA NP administration, 231 VEGF tumors have lower Chkα protein than 231 WT tumors. (**D**) Representative in vivo color-coded total choline intensity map of a 231 VEGF tumor overlaid with T_1_ weighted MR image, obtained before treatment and at 24 h after the second dose of Chkα siRNA NPs.

**Figure 6 pharmaceutics-14-01260-f006:**
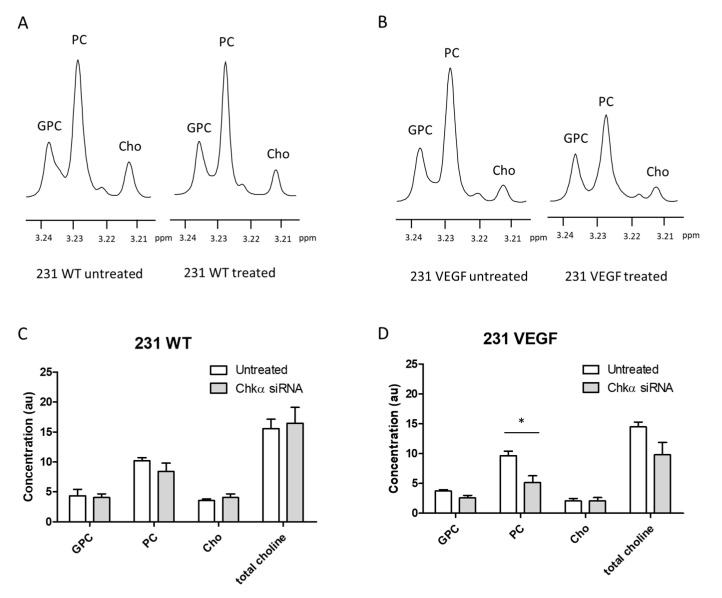
Representative ^1^H MR spectra expanded to display choline metabolite region obtained from 231 WT (**A**) and 231 VEGF (**B**) tumor extracts with and without Chkα siRNA NP administration. Quantitative analysis (arbitrary unit, au) of ^1^H MR spectra from 231 WT (**C**) and 231 VEGF (**D**) tumor extracts. 231 WT tumors showed no significant difference in choline metabolites following siRNA PEG-PEI NP administration. PC levels significantly decreased in 231 VEGF tumors following siRNA PEG-PEI NP administration. Values represent mean ± SEM, * *p* ≤ 0.05, *n* = 8 each for treated 231 WT or 231 VEGF tumors, *n* = 3 each for control 231 WT or 231 VEGF tumors. GPC: glycerophosphocholine, PC: phosphocholine, Cho: choline, and total choline: Cho + PC + GPC.

## Data Availability

All data supporting the results and conclusions are available in the main text or in the Appendix A. The raw data supporting the conclusions of this article will be made available by the authors.

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
