# Peer review of "VEGF Overexpression Significantly Increases Nanoparticle-Mediated siRNA Delivery and Target-Gene Downregulation"

_pharmaceutics, 2022, doi:10.3390/pharmaceutics14061260_

Round 1
Reviewer 1 Report
PEI is known to be associated with severe cytotoxicity, have the authors conducted any long-term cytotoxicity with PEI? Also, they need to mention the Physicochemical characterization such as Zeta and PDI, it was mentioned that the proposed NP tends to aggregate with high PDI in some cases but what are the PDIs,?
From both the in vivo and ex vivo images the NP tends to accumulate mainly in the liver and kidney which indicate high clearance of the NP, have the authors conducted hepato, and nephrotoxicity after long exposure, what is the clearance rate of these NPs?
Related to the previous Q., the NPs tend to accumulate in the spleen in the case of WT compared to VEGF, can the authors explain why it is the case?
For Figures 6 C and D, the baseline Cho levels seem to be < 2 au compared to 5 au in WT, can the author confirm this with statistical analysis to eliminate any other factor?
Author Response
- PEI is known to be associated with severe cytotoxicity, have the authors conducted any long-term cytotoxicity with PEI? Also, they need to mention the Physicochemical characterization such as Zeta and PDI, it was mentioned that the proposed NP tends to aggregate with high PDI in some cases but what are the PDIs?
Response: Yes we agree. Polyethylene glycol modification of PEI, such as used in this study, significantly reduces toxicity. We have previously evaluated PEG-PEI cytotoxicity using an MTT assay in MDA-MB-231 cells treated for 4 days with varying concentrations of PEG-PEI and found that up to 0.3 M PEG-PEI concentrations did not reduce cell viability (Li et al., ACS Nano, 2010). Based on the injection dose of 0.1 mg PEG-PEI in a 25 gm mouse, we anticipate that the in vivo concentration of PEG-PEI of ~0.1 M did not induce cytotoxicity over a 4 day period by which time the complex would have been excreted. The zeta-potential of the PEG-PEI siRNA complex is approximately 7.0 mV as previously estimated (Li et al., ACS Nano, 2010), and the PDI is 0.336. These points have been included in the revised manuscript on Page 3, lines 118-126 and Page 12, lines 392-397.
- From both the in vivo and ex vivo images the NP tends to accumulate mainly in the liver and kidney which indicate high clearance of the NP, have the authors conducted hepato, and nephrotoxicity after long exposure, what is the clearance rate of these NPs?
Response: Because our primary goal was to understand the role of the tumor microenvironment, specifically tumor VEGF expression, on NP delivery, we did not perform studies characterizing long term effects of the NPs on hepato- and nephrotoxicity, but these studies should be performed in the future.
- Related to the previous Q., the NPs tend to accumulate in the spleen in the case of WT compared to VEGF, can the authors explain why it is the case?
Response: In the representative image shown in Figure 4A, NP accumulation was higher in the WT compared to the VEGF tumor bearing mice. However, when averaged across all the mice in each group there was no significant difference in spleen accumulation between the two groups. We have clarified this in the revised manuscript on Page 7, lines 282-285.
- For Figures 6 C and D, the baseline Cho levels seem to be < 2 au compared to 5 au in WT, can the author confirm this with statistical analysis to eliminate any other factor?
Response: Thank you for noticing this. We performed statistical analysis of the baseline Cho levels in the WT (3.6 ± 0.49 au) and VEGF group (2.14 ± 0.79 au) but the p-value was not statistically significant. We have clarified this in the revised manuscript on Page 10, lines 343-347.

Reviewer 2 Report
The manuscript pharmaceutics-1706226 VEGF overexpression significantly increases nanoparticle-mediated siRNA delivery and target-gene downregulation by Shanshan Tan et al. describes the role of VEGF overexpression in the tumor site in improving siRNA delivery using theranostic nanoparticles with imaging reporters.
The manuscript is logical and well written. The prospects for the practical use of the results obtained are high. The paper will definitely be of interest to the readers of Pharmaceutics.
Questions and comments:
1) Line 99 - 1H NMR spectrum should be presented in the supplementary materials.
2) The authors report the characteristics of nanoparticles based on PEG-modified PEI (hydrodynamic radius of 10 nm). However, the size of miRNA-loaded nanoparticles is also important, since the gene delivery is significantly affected by the size of the nanocontainers, e.g., due to the EPR-effect.
Author Response
Questions and comments:
- Line 99 - 1H NMR spectrum should be presented in the supplementary materials.
Response: We have included a 1H NMR spectrum in Figure S1 of the revised manuscript.
- The authors report the characteristics of nanoparticles based on PEG-modified PEI (hydrodynamic radius of 10 nm). However, the size of miRNA-loaded nanoparticles is also important, since the gene delivery is significantly affected by the size of the nanocontainers, e.g., due to the EPR-effect.
Response: We have included the TEM data in Figure S2 and the DLS data in Figure S3 that evaluate the size distribution of the NPs in the revised manuscript. The TEM data identified the particle diameter as 15.4 ± 3.8 nm. The hydrodynamic diameter of the NPs in solution was 108 nm. The Poly-disperse Index (PdI) was 0.336. We have included this information on Page 3, lines 118-126.

Reviewer 3 Report
The research is well designed and executed, but it lacks key details on the Chkα siRNA PEI NPs. Below are some of the comments;
- Have you tested the NPs on any other VEGF overexpressing xenografts, except triple-negative MDA-MB-231 human breast cancer xenografts? If not, why? Do you know NPs react differently based on the protein corona of cells?
- Add the DLS data to the main manuscript? What's the zeta-potential and stability of the Chkα siRNA PEI NPs? Have you performed any studies on it?
- Why the other characterizations (TEM, FTIR, XPS) of Chkα siRNA PEI NPs is not performed? What's the structure of Chkα siRNA PEI NPs?
- Have you tested the Chkα siRNA PEI NPs for the protein corona (PC) in-vitro? Do you know that PC can significantly impact the application of NPs?
- What's the ADME "absorption, distribution, metabolism, and excretion" Profile of Chkα siRNA PEI NPs? Can you explain how it is metabolised, and excreted in-vivo?
- Ex vivo organ biodistribution of Chkα siRNA NPs (Figure 4) shows high intensity in the kidneys, what's the reason that the kidneys received more NPs than the liver?
You need to include the following data in the manuscript;
- TEM/SEM image of Chkα siRNA PEI NPs.
- DLS data of Chkα siRNA PEI NPs.
Regarding the PC and ADME, you can add more discussion and references from the literature.
Indeed delivering the siRNA via NPs is a promising technique for targeted gene regulations but the nano-bio interface should be fully explored.
Author Response
The research is well designed and executed, but it lacks key details on the Chkα siRNA PEI NPs. Below are some of the comments;
Response: Thank you for this comment.
- Have you tested the NPs on any other VEGF overexpressing xenografts, except triple-negative MDA-MB-231 human breast cancer xenografts? If not, why? Do you know NPs react differently based on the protein corona of cells?
Response: Thank you for this comment and for identifying the possibility that NPs can react differently based on the protein corona of cells. In our study, we used a pair of isogenic cancer cell lines with and without VEGF expression to investigate the effects of increasing tumor vascularity on NP delivery and downregulation of the target gene. Because we used isogenic cancer cells in SCID mice we do not anticipate that plasma proteins will be altered, but this effect should also be considered. We have included this and the corresponding reference (Ovais et al., Nanomedicine, 2020) on protein corona in the discussion on Page 12, lines 398-402.
- Add the DLS data to the main manuscript? What's the zeta-potential and stability of the Chkα siRNA PEI NPs? Have you performed any studies on it?
Response: We have included DLS data in supplementary Figure S2 in the revised manuscript. The zeta-potential of the PEG-PEI siRNA complex is approximately 7.0 mV as previously estimated (Li et al., ACS Nano, 2010). We have not performed characterization of stability in this study, but previous studies have demonstrated the stability of PEG-PEI NPs in serum (Merkel et al., J. Controlled Release, 2009). We have included these points in the revised manuscript on Page 3, lines 118-126.
- Why the other characterizations (TEM, FTIR, XPS) of Chkα siRNA PEI NPs is not performed? What's the structure of Chkα siRNA PEI NPs?
Response: We have included TEM and DLS data in supplementary Figures S1 and S2 in the revised manuscript and have provided additional details in the section ‘Preparation and characterization of Chkα’ siRNA PEG-PEI NPs on Page 3, lines 102-126.
- Have you tested the Chkα siRNA PEI NPs for the protein corona (PC) in-vitro? Do you know that PC can significantly impact the application of NPs?
Response: Please see our response to Comment 1.
- What's the ADME "absorption, distribution, metabolism, and excretion" Profile of Chkα siRNA PEI NPs? Can you explain how it is metabolised, and excreted in-vivo?
Response: The reviewer makes an excellent suggestion. Because of the size of the NPs used in this study, we anticipate that they were cleared by the reticuloendothelial system (RES). Several studies that have investigated the pharmacokinetics and clearance of PEG-PEI NPs (e.g. Malek et al., Toxicology and Applied Pharmacology, 2009). These NPs are cleared by the reticuloendothelial system (RES), with degradation products that undergo renal and hepatobiliary elimination (Poon et al., ACS Nano, 2019). Our imaging studies showed the highest uptake in the kidney followed by the liver 24 h after two doses of the NPs. We have included these points and references in the revised manuscript on Page 12, lines 402-410.
- Ex vivo organ biodistribution of Chkα siRNA NPs (Figure 4) shows high intensity in the kidneys, what's the reason that the kidneys received more NPs than the liver?
Response: Our images were acquired 24 h after the second dose of the PEG-PEI NPs. As mentioned in the earlier response, by 24 h degradation of the NPs will undergo both renal and hepatobiliary elimination. The images suggest that there was more renal clearance at 24 h. We have included this point in the revised manuscript on Page 12, lines 402-410.
You need to include the following data in the manuscript;
- TEM/SEM image of Chkα siRNA PEI NPs.
- DLS data of Chkα siRNA PEI NPs.
Response: We have provided this in the revised manuscript. Please see response to earlier comments.
Regarding the PC and ADME, you can add more discussion and references from the literature.
Response: We have included this in the revised manuscript. Please see response to earlier comments.
Indeed delivering the siRNA via NPs is a promising technique for targeted gene regulations but the nano-bio interface should be fully explored.
Response: Thank you for this comment.

Reviewer 4 Report
see the attached file

Author Response
- All abbreviations should be first identified before use them even if they were in abstract or another part of the manuscript
Response: We have corrected this in the revised manuscript.
- The abstract does not report the main findings of the study in a clear manner. For example general expressions are used which do not provide useful information to the readers. Information that is more specific is required in the abstract.
Response: We have provided specific information in the abstract. Please see Page 1, lines 28-29.
- Key words must be arranged alphabetically
Response: Thank you for identifying this. We have corrected this in the revised manuscript.
- The introduction does not point out the gap of the literature. The study seeks to fill and novelty of the study over the existing literature. This point showed be further elaborated.
Response: We have further elaborated this point in the revised manuscript. Please see Page 2, lines 46-50.
- The determination of protein need a reference (lines 192-198)
Response: We have included a reference in the revised manuscript (Subedi et al., Analytical Biochemistry, 2019).
- Why the number of samples is differ (231 WT (n=4) and 231 VEGF (n=5) tumors)
Response: For the data in Figure 1, unfortunately, 1 animal died in the WT group prior to tumor growth or treatment.
- A relevant hypothesis for the study is missing from the introduction. A true scientific question should be formed.
Response: We have clarified our scientific question in the introduction of the revised manuscript. Please see Page 2, lines 46-50.
- Enhance the resolution of figure 2
Response: We have improved the resolution of Figure 2 (Shanshan please make sure this is done).
- Simplify the statement in the paper. Please combine and condense the discussion and conclusion.
Response: We have combined the discussion and conclusion in the revised manuscript.
- The conclusion section is missing.
Response: Please see response to Comment 9.

Round 2
Reviewer 1 Report
Authors have implemented and responded to comments in revised manuscript
Reviewer 3 Report
Thanks for addressing the comments!
Reviewer 4 Report
Dear Editor Journal of Pharmaceutics
Manuscript ID: pharmaceutics-1706226
I re-reviewed the manuscript “VEGF overexpression significantly increases nanoparticle-me- diated siRNA delivery and target-gene downregulation” again and the authors made all the amendments that I asked before so I think the manuscript is suitable for publishing
This manuscript is a resubmission of an earlier submission. The following is a list of the peer review reports and author responses from that submission.